# Differential Sensitivity of Germline and Somatic *BRCA* Variants to PARP Inhibitor in High-Grade Serous Ovarian Cancer

**DOI:** 10.3390/ijms241814181

**Published:** 2023-09-16

**Authors:** Julie A. Vendrell, Iulian O. Ban, Isabelle Solassol, Patricia Audran, Simon Cabello-Aguilar, Delphine Topart, Clothilde Lindet-Bourgeois, Pierre-Emmanuel Colombo, Eric Legouffe, Véronique D’Hondt, Michel Fabbro, Jérôme Solassol

**Affiliations:** 1Laboratoire de Biologie des Tumeurs Solides, Département de Pathologie et Oncobiologie, CHU Montpellier, Université de Montpellier, 34295 Montpellier, France; j-vendrell@chu-montpellier.fr (J.A.V.); iulian.ban@chu-montpellier.fr (I.O.B.); s-cabelloaguilar@chu-montpellier.fr (S.C.-A.); 2Unité de Recherche Translationnelle, Institut Régional du Cancer de Montpellier (ICM), 34090 Montpellier, France; isabelle.solassol@icm.unicancer.fr; 3Département d’Anatomo-Pathologie, Institut Régional du Cancer de Montpellier (ICM), Université de Montpellier, 34090 Montpellier, France; p-audran@chu-montpellier.fr; 4Montpellier BioInformatics for Clinical Diagnosis (MOBIDIC), Molecular Medicine and Genomics Platform (PMMG), CHU Montpellier, 34295 Montpellier, France; 5Oncologie Médicale, CHU Montpellier, Université de Montpellier, 34295 Montpellier, France; d-topart@chu-montpellier.fr (D.T.); c-lindetbourgeois@chu-montpellier.fr (C.L.-B.); 6Département de Chirurgie Oncologique, Institut Régional du Cancer de Montpellier (ICM), 34090 Montpellier, France; pierre-emmanuel.colombo@icm.unicancer.fr; 7Oncologie Médicale, Institut de Cancérologie du Gard, 30900 Nîmes, France; e.legouffe@oncogard.com; 8Département d’Oncologie Médicale, Institut Régional du Cancer de Montpellier (ICM), Université de Montpellier, 34090 Montpellier, France; veronique.dhondt@icm.unicancer.fr (V.D.); michel.fabbro@icm.unicancer.fr (M.F.); 9Montpellier Research Cancer Institute (IRCM), Institut National de la Santé et de la Recherche Médicale (INSERM) U1194, University of Montpellier, 34298 Montpellier, France

**Keywords:** ovarian cancer, *BRCA1* and *BRCA2* alteration, germline and somatic variant, PARP inhibitor

## Abstract

Purpose: The introduction of PARP inhibitors (PARPis) as a treatment option for patients with high-grade serous ovarian cancer (HGSOC) modified the approach of *BRCA* testing worldwide. In this study, we aim to evaluate the impact of *BRCA1* and *BRCA2* variants on treatment response and survival outcomes in patients diagnosed in our institution. Methods: A total of 805 HGSOC samples underwent *BRCA1* and *BRCA2* variant detection by using next-generation sequencing (NGS). Among them, a pathogenic alteration was detected in 104 specimens. Clinicopathological features and germline status were recovered, and alteration types were further characterized. The clinical significance of variant type in terms of response to chemotherapy and to PARPis as well as overall survival were evaluated using univariate analysis. Results: In our cohort, 13.2% of the HGSOC samples harbored a pathogenic *BRCA1* or *BRCA2* variant, among which 58.7% were inherited. No difference was observed between germline and somatic variants in terms of the gene altered. Interestingly, patients with somatic variants only (no germline) demonstrated better outcomes under PARPi treatment compared to those with germline ones. Conclusion: The determination of the inheritance or acquisition of *BRCA1* and *BRCA2* alterations could provide valuable information for improving management strategies and predicting the outcome of patients with HGSOC.

## 1. Introduction

High-grade serous ovarian cancer (HGSOC) is the most prevalent type of ovarian cancer, accounting for 70% of ovarian tumors [1]. This highly aggressive cancer represents a major clinical challenge as its indolent nature means that it is often diagnosed at an advanced stage [2]. Approximately 84% of patients are diagnosed with advanced, stage III-IV, disease according to the International Federation of Gynecology and Obstetrics (FIGO) staging system [1]. Such late diagnosis significantly impacts prognosis, with a 5-year overall survival (OS) rate of approximately 10–30% for stages III–IV compared to 85–90% for stages I–II [3]. The standard of care for HGOSC associates cytoreductive surgery and a combined platinum and taxane chemotherapy. This first-line chemotherapy produces an excellent 70% response rate. However, despite an aggressive first-line treatment, 75% of patients will relapse within 3 years and eventually die from the disease [4].

Poly-ADP-ribose polymerase inhibitors (PARPis) represent a recent significant breakthrough in targeted therapy for HGSOC. They are particularly effective in tumors with homologous recombination DNA repair deficiency (HRD). In these tumors, PARP inhibition leads to synthetic lethality from the combination of two non-lethal events (PARP inhibition and HRD) [5,6]. The inability of cancer cells to repair DNA damage leads to the accumulation of DNA breaks and cell death. However, some patients relapse due to acquired mechanisms of resistance [7,8]. Several studies suggested that PARPi sensitivity may vary based on the tumor’s genetic and molecular features, implying that multiple factors could contribute to the variability in the clinical response to PARP therapies [9,10,11,12].

PARPis have received FDA-approval for the treatment of HGSOC patients with *BRCA1* or *BRCA2* alteration (germline and/or somatic), following response to first-line platinum-based chemotherapy [13]. According to various studies, around 16–23% of the HGSOC samples harbor a pathogenic variant on *BRCA1* or *BRCA2* at the time of diagnosis [14,15,16]. While the majority of these alterations can be detected at the germline level in patients with hereditary breast and ovarian cancer syndrome, almost 30% of these changes occur in the tumor itself and can only be identified at the somatic level [17]. These findings highlight the importance of identifying the presence of pathogenic *BRCA1* or *BRCA2* variants in tumor tissue in order to evaluate the indication for PARPis in HGSOC patients.

Although there has been a substantial amount of research conducted in this area, our current understanding of the link between the origin of *BRCA1* and *BRCA2* alteration (inherited or acquired) and the patient’s outcome remains unclear. To address this gap, we conducted a retrospective analysis of HGSOC tumors diagnosed in our institution, aiming to characterize the somatic or germline origin of any *BRCA1* and *BRCA2* alterations while assessing its impact on sensitivity to chemotherapy or PARPis and on outcome.

## 2. Results

### 2.1. Patient Cohort and Alteration Descriptions

From 2016 to 2021, the Pathology Laboratory at the University Hospital of Montpellier (France), received 805 formalin-fixed, paraffin-embedded (FFPE) samples from HGSOC patients for full exon screening of *BRCA1* and *BRCA2* by using next-generation sequencing (NGS). Out of the 787 samples eligible for analysis, 104 (13.2%) harbored a pathogenic/likely pathogenic (pathogenic) variant, 40 (5.1%) harbored a variant of uncertain significance (VUS) and 643 (81.7%) were wildtype (WT) for *BRCA1* and *BRCA2* (Figure 1 and Figure 2A). Among the samples with a pathogenic alteration, four also presented a VUS on the same gene (two cases for *BRCA1* and two cases for *BRCA2*) and four had a VUS on the other gene (two cases with a pathogenic variant *BRCA1* and a VUS on *BRCA2* and two cases with a pathogenic variant on *BRCA2* and a VUS on *BRCA1*).

Among the pathogenic variants, 69 (66.3%) alterations affected *BRCA1* and 35 (33.7%) affected *BRCA2* (Appendix A). Small insertions/deletions (indels) were the most frequent DNA alterations detected, affecting 68.1% of *BRCA1* and 74.3% of *BRCA2* genes (Appendix A). Variants affecting splicing were detected in three cases each for *BRCA1* and *BRCA2*, whereas missense variants were only detected in four *BRCA1* altered specimens, with one variant disrupting the translation initiation codon (Appendix A, Figure 2B and Appendix A). Additionally, 86.9% and 91.4% of these alterations induced the appearance of a premature termination codon due to frameshift or nonsense variants in *BRCA1* and *BRCA2*, respectively (Figure 2B and Appendix A). Although the mutations seemed to be distributed throughout the genes, seven variants on *BRCA1* were detected in more than one sample, with one variant reported in three samples (p.Glu1005Asnfs*19, Appendix A).

### 2.2. Clinicopathological Characteristics of the Patients with a Pathogenic Variant

Clinical and pathological features from patients harboring pathogenic variants are presented in Table 1. Patients in our cohort had a median age at diagnosis of 63.4 years (range: 30.4–87.0). Among those with available FIGO staging, 85.9% had stage III and IV disease. In accordance with international guidelines, women with a pathogenic alteration of *BRCA1* or *BRCA2* were advised to undergo genetic counseling. Blood sample testing performed for 80 of the 104 patients allowed for the classification of variants identified on the tumor samples into two classes: (i) the somatic variants detected in the tumor sample only and (ii) the germline variants identified both in the blood and the tumor samples (Figure 3A and Appendix A). Germline status was not available for the remaining 24 samples showing pathogenic variants. In our cohort, 47 alterations were classified as germline (58.7%) and 33 were classified as somatic (41.2%), with a similar distribution between *BRCA1* and *BRCA2* (*p* = 0.68, Figure 3A and Appendix A). There was no significant difference regarding the age at diagnosis and the stage (Appendix A) in both groups. Moreover, the distribution of alterations throughout the genes indicated no apparent accumulation in specific regions (Figure 3B,C).

### 2.3. Clinical Outcome of Patients According to the Presence of Germline or Somatic Pathogenic Alterations

Among the 80 patients who underwent genetic testing, complete clinical follow-up was available for 66. All had received first-line platinum-based chemotherapy treatment leading to relapse in 53. Among these latter, 49 received PARPis as maintenance treatment after demonstrated sensitivity to platinum-based chemotherapy. We next assessed the prognostic value of the variant status (*BRCA1* vs. *BRCA2* and germline vs. somatic) with regards to response to treatments and overall survival. *BRCA1*- and *BRCA2*-altered patients showed similar responses to platinum therapy and PARP inhibitor treatment as well as showed similar overall survival (Table 2 and Figure 4A–C). Interestingly, patients with a somatic variant without corresponding germline mutation showed a better outcome than those with a germline alteration (*p* = 0.049; log-rank test) under PARPi treatment, whereas no differences were observed under platinum therapy (Table 2 and Figure 4D,E). In addition, women with a somatic alteration also appeared to survive longer than those with an inherited alteration (Figure 4F, 5-year survival rate of 86.4% and 63.7%, respectively) although the difference did not reach significance (*p* = 0.17).

## 3. Discussion

The introduction of PARPis as a treatment option for patients with HGSOC with a pathogenic variant in *BRCA1* or *BRCA2* has led to significant changes in the approach to BRCA testing. Initially used to identify patients with a predisposition to develop hereditary breast–ovarian cancer (HBOC) syndrome, germline *BRCA* testing is now also essential to identify patients most likely to benefit from PARPis. Interestingly, the first clinical studies suggested that patients with a somatic *BRCA* alteration may also respond to PARPis [2,18,19]. The SOLO-1 study showed that maintenance therapy with olaparib improved PFS for patients with deleterious germline or somatic *BRCA* alteration [2]. The phase 3 PAOLA-1 trial showed improved PFS and OS with olaparib plus bevacizumab compared to bevacizumab alone for patients with *BRCA* deleterious alteration or HRD-positive tumors [20,21]. Following the SOLO-1 findings, olaparib was approved by the FDA in 2018 for first-line maintenance treatment in germline and/or somatic *BRCA*-altered advanced HGSOC. Debates still remain regarding the administration, dosages, timing and scheduling of bevacizumab [22]. More recently, rucaparib and niraparib have been approved in first-line maintenance treatment for patients with *BRCA* alteration and HRD-positive tumors [23,24]. Consequently, tumor *BRCA* testing allowing for the rapid identification of both germline and somatic has been implemented in clinical practice. Several studies reported that 16–23% of the HGSOC samples harbored a pathogenic variant on *BRCA1* or *BRCA2* at diagnosis, with around 28–39% of these alterations present only at the somatic level [25,26,27]. In our study, we detected a slightly lower percentage of pathogenic variants (13%) and a higher proportion of somatic alterations (44%). This is explained by the fact that, for some patients, germline *BRCA* testing had already been performed in a first attempt early in diagnosis and, because no alteration had been detected, the tumor samples of these patients were therefore sent to our laboratory for further analysis. Thus, recent data and our present study support essential tumor testing in patients even without germline alteration to identify those with a somatic variant in whom PARPi treatment would also be beneficial. Nonetheless, a growing body of evidence suggests that PARPis also confer benefits in the management of advanced HGSOC regardless of the patient’s molecular *BRCA* status [28]. This expands the range of potential therapeutic options for individuals with ovarian cancer.

In our study, we found germline and somatic alterations throughout both *BRCA1* and *BRCA2* genes with no obvious regional accumulation. Likewise, alteration type was not more frequent in one gene compared to the other. Alterations led to the appearance of a premature termination codon in 89.4% of cases, producing an altered mRNA that is degraded by the nonsense-mediated mRNA decay pathway [29]. Missense alterations were located in key functional domains of BRCA1, such as the RING domain. Other reported variants affected splicing sites and their impact on the protein remains uncharacterized. Recently, two studies suggested a disparity in the sensitivity of the cells to cisplatin and/or PARPis based on the *BRCA* variant involved [30,31]. Wang et al. demonstrated that cell lines and tumor samples harboring a frameshift variant in the exon 10 of *BRCA1* expressed a *BRCA1* isoform that confers a partial resistance to chemotherapy and PARPis. However, the authors also reported that not all the variants led to the same response to treatment and that patient-derived xenograft models established from two patients harboring the same variant responded differently to olaparib treatment [30]. Further work is therefore still needed to better identify women who will benefit from PARPis according to the alteration detected.

While *BRCA*-altered patients are now largely known to respond better to platinum-based chemotherapy and PARPis [32], the impact of the altered gene or the origin of the alteration (inherited or acquired) on clinical outcomes remains unclear. While some studies suggested an improved response to chemotherapy and improved survival for *BRCA2*-altered patients [33,34], more recent studies failed to detect any difference between patients with *BRCA1* or *BRCA2* variants [7,35]. Patients in our cohort with *BRCA1* or *BRCA2* variants exhibited comparable outcomes in terms of their response to platinum therapy and PARPis as well as their demonstrated overall survival. Furthermore, we found no significant difference between patients with somatic or germline alterations in terms of response to chemotherapy, as previously reported [36]. Interestingly, however, when exploring the impact of somatic or germline alteration on response to PARPis for the first time, we found a better outcome in patients with a somatic variant. Moreover, women with a somatic alteration also appeared to survive longer than those with an inherited alteration. Several hypotheses can be proposed to elucidate these observations. A plausible explanation is that treatments might impact non-tumoral cells harboring germline alterations. Additionally, germline alterations in genes associated with DNA repair pathways could lead to heightened genomic instability and the emergence of somatic alterations, consequently influencing patient outcomes. Recent findings have also revealed that germline alterations may have an impact on the tumor microenvironment and influence the immune response within tumors [37]. These may provide potential explanations for the observed differences in survival outcomes between patients with germline alterations and those with somatic alterations. Additional research efforts are required to enhance our understanding of this highly complex issue.

The small size of samples and the retrospective nature of our study however give us reason to be cautious about the impact of these findings and the need for validation by using a larger cohort. Distinguishing patients harboring germline from those with somatic alterations may prove useful in management decisions that would allow for more rigorous monitoring of those with a germline alteration at risk of earlier relapse under PARPis.

In summary, our results indicate a beneficial impact of determining the origin (germline or somatic) of *BRCA* alterations on the accurate assessment of patient prognosis. They also highlight the importance of multidisciplinary collaboration amongst oncologists, biologists and geneticists in the management of women with HGSOC.

## 4. Material and Methods

### 4.1. Patients and Samples Collection

From 2016 to 2021, 805 samples from patients with peritoneal, ovarian and fallopian tube carcinoma underwent somatic *BRCA1* and *BRCA2* variant detection analysis at the University Hospital of Montpellier (Montpellier, France). Patients were included in the cohort if they were above 18 years old, had a confirmed diagnosis of HGSOC, had approved informed consent statement and if their tissue was available and analyzable. Individuals were excluded if they had a history of other hereditary syndromes or were undergoing treatment for another malignancy. The study enrolled 787 patients. Approval was obtained from the Institutional Review Board of the Montpellier University Hospital. All corresponding lesions were excised and submitted for standard pathologic examination. The percentage of tumor cells in the series ranged from 20% to 100%. Tissue punches using a 1 mm needle or macro-dissection of 10 µm thick sections were performed on FFPE tumor blocks to increase the percentage of tumor cells in the sample. Information regarding clinicopathological data and genetic testing was retrieved from the medical records of the 104 patients with a pathogenic variant.

### 4.2. DNA Extraction and Qualification

Genomic DNA was extracted using the Maxwell^®^ RSC DNA FFPE Kit (Promega, Madison, WI, USA) according to the manufacturer’s recommendations. Extracted DNA was quantified using the Qubit dsDNA Broad Range Assay Kit in combination with a Qubit fluorimeter (Thermo Fisher Scientific, Waltham, MA, USA). DNA integrity of the samples was assessed using the QC Plex Kit following the manufacturer’s instructions (Agilent Technologies, Santa Clara, CA, USA). Briefly, this assay is based on a multiplex PCR that generates 7 PCR fragments with several lengths ranging from 100 to 700 bp. PCR products are then analyzed using D1000 ScreenTapes in combination with a 4200 TapeStation instrument (Agilent Technologies). After analysis, the areas of the relevant peaks are uploaded to the ΔQC calculator (Agilent Technologies) to determine the DNA quality coefficient (ΔQC).

### 4.3. BRCA1 and BRCA2 Variants Analysis

Library preparation was performed using the BRCA Master Plus Dx Kit following the manufacturer’s instructions (Agilent Technologies). Briefly, the recommended DNA input volume to use per reaction is determined by the ΔQC calculated per sample and amplified using a target-specific multiplex PCR run (four different primer pools per sample). A second PCR run was then performed to incorporate molecular barcodes and sequencing adaptors. The PCR products were then purified with Agencourt AMPure XP (Beckman Coulter, Nyon, Switzerland), quantified on a Qubit instrument and pooled to equimolar concentration. The library was paired-end sequenced (2 × 250 cycles) on a MiSeq instrument (Illumina, San Diego, CA, USA).

### 4.4. Bioinformatics and Data Analysis

The FASTQ files were analyzed with the SeqNext software version 4.3.1 (JSI Medical Systems, Ettenheim, Germany) using the following criteria: (i) reads for which 75% of the bases have a quality score below Q20 were excluded from the analysis; (ii) only amplicons with a read depth of 300× or greater were used for variant calling; (iii) variants were called if they were present in at least 5 reads. The *BRCA1* (NM_007294.2) and *BRCA2* (NM_000059.3) sequences from the National Center for Biotechnology Information database were used as references. Variants reported were annotated following the American College of Medical Genetics (ACMG) recommendations and classified using germline public databases as references. Thus, variants reported as pathogenic or likely pathogenic in the *BRCA* share, *BRCA*-UMD or *BRCA* Exchange databases were considered to be pathogenic (class 5). Variants for which the clinical impact had not been clearly demonstrated or with conflicting evidence for benign or pathogenic impact in the mentioned database (class 3) were considered to be VUS. Otherwise, the sample was assigned to the WT *BRCA1* and *BRCA2* group (class 1) (Appendix A).

### 4.5. Statistical Analysis

The significance of the differences between survival rates was estimated using the Kaplan–Meier method and was ascertained with the log-rank test using SPSS^®^ software version 21 (SPSS Inc., Armonk, NY, USA). A *p* value < 0.05 was considered significant.

## Figures and Tables

**Figure 1 ijms-24-14181-f001:**
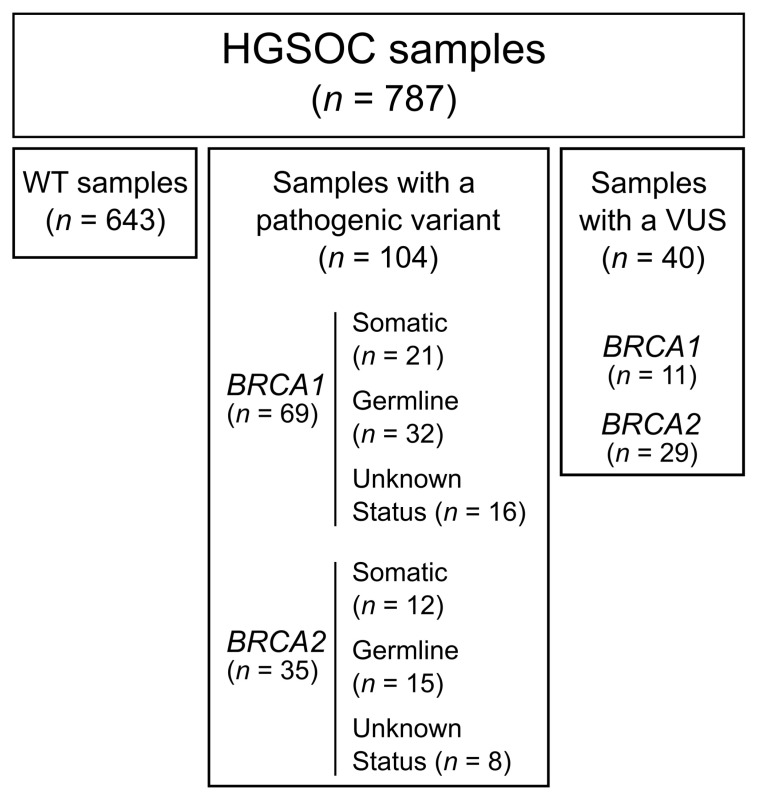
Flow diagram of the tumor samples present in the cohort. VUS, variant of uncertain significance.

**Figure 2 ijms-24-14181-f002:**
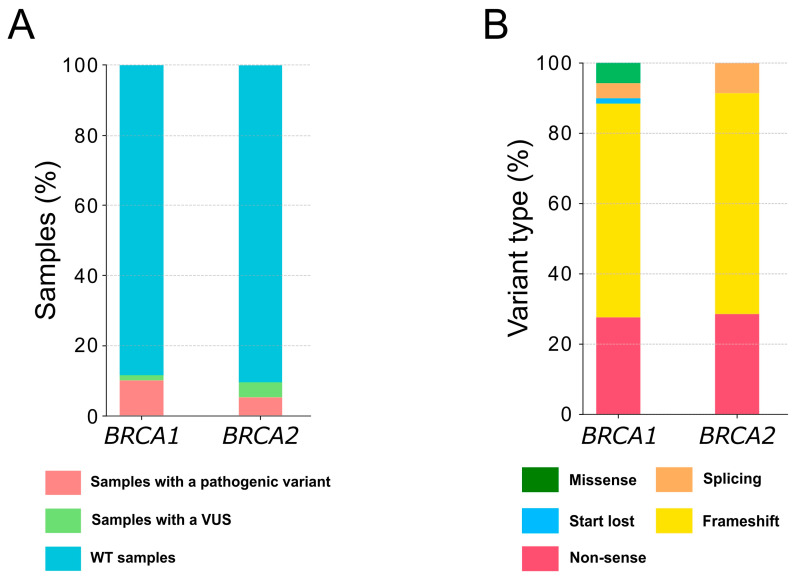
Representativeness of the *BRCA1* and *BRCA2* variants detected in the cohort. (**A**) Class of variants identified in *BRCA1* and *BRCA2*; (**B**) type of pathogenic alterations reported in *BRCA1* and *BRCA2*. VUS, variant of uncertain significance.

**Figure 3 ijms-24-14181-f003:**
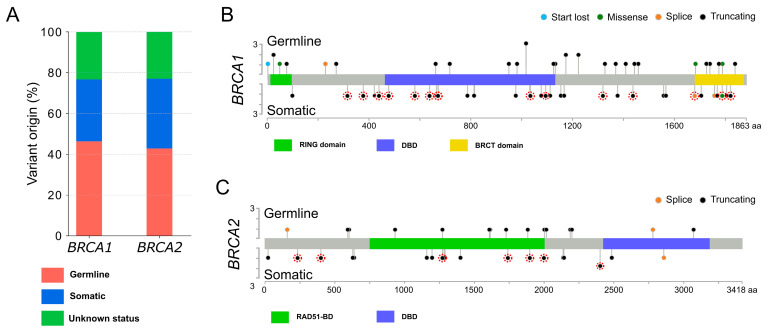
Landscape of the pathogenic *BRCA1* and *BRCA2* alterations detected. (**A**) Proportion of germline and somatic variants observed in *BRCA1* and *BRCA2*; (**B**) *BRCA1* and (**C**) *BRCA2* mutation plots illustrating the variant localization along the protein sequence. Colored boxes indicate distinct functional domains. Colored dots donate various variant types: blue for disruption of the translation initiation codon, green for missense variants, orange for splice variants and black for the emergence of a premature termination codon. Germline variants are placed above the protein sequence, while somatic variants are positioned below. Variants with unknown germline status are placed below and encircled in red. Mutation plots were generated using the cBioPortal MutationMapper tool. RING domain, really interesting new gene domain; DBD, DNA-binding domain; BRCT domain, C-terminal domain of BRCA1, RAD51-BD, RAD51 binding domain.

**Figure 4 ijms-24-14181-f004:**
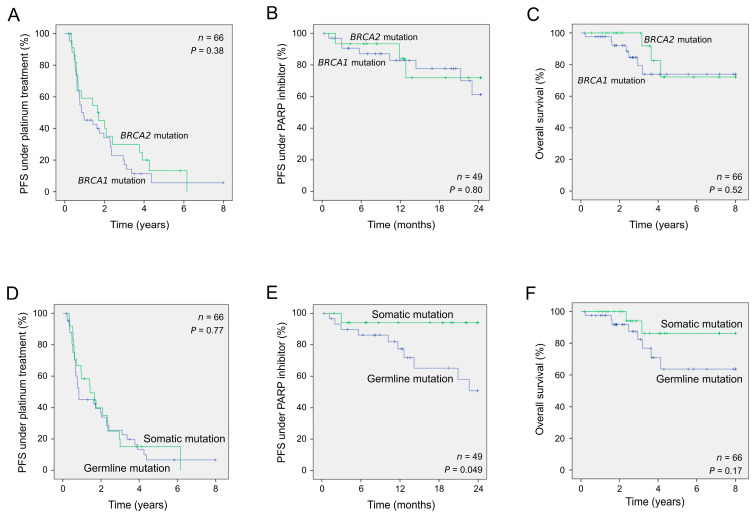
Clinical outcomes of patients according to *BRCA1* or *BRCA2* variant types. Kaplan–Meier analyses of patients harboring pathogenic variants based on the specific gene altered (**A**–**C**) and its germline or somatic status (**D**–**F**). PFS, progression-free survival.

**Table 1 ijms-24-14181-t001:** Patient and specimen characteristics (*n* = 104).

		Patients (*n* = 104)
Age at diagnosis		
	Median (range)	63.4 (30.4–87.0)
	<60	41
	≥60	63
FIGO Stage		
	I	6
	II	3
	III	36
	IV	19
	Unknown	40
Type of specimen		
	Biopsy	28
	Surgical specimen	70
	Unknown	6
Tumor cell content		
	<50%	16
	≥50%	85
	Unknown	3
Clinical follow-up		
	Yes	91
	No	13
PARP inhibitor administration		
	Yes	58
	No	33
	Unknown	13
Germline status		
	Known	80
	Unknown	24
Mutation present		
	At the germline level	47
	At the somatic level only	33
	Genetic result not available	24
Gene mutated		
	*BRCA1*	69
	*BRCA2*	35

**Table 2 ijms-24-14181-t002:** Clinical outcome according to the *BRCA1* and *BRCA2* variant types.

		Number of Samples	Univariate Analysis		
			HR	95% CI	*p* ^a^
PFS to first-line platinum-based chemotherapy					
	Gene affected (*BRCA1*; *BRCA2*)	66	1.30	0.74–2.27	0.38
	*BRCA* genetic status (germline; somatic)	66	1.09	0.63–1.91	0.77
PFS to second-line PARP inhibitor treatment					
	Gene affected (*BRCA1*; *BRCA2*)	49	1.17	0.32–4.24	0.80
	*BRCA* genetic status (germline; somatic)	49	3.42	1.01–11.63	0.049
OS					
	Gene affected (*BRCA1*; *BRCA2*)	66	1.51	0.43–5.39	0.52
	*BRCA* genetic status (germline; somatic)	66	2.40	0.68–8.48	0.17

^a^ *p* value (log-rank test) was considered significant when *p* < 0.05; PFS: progression-free survival; OS: overall survival; HR: hazard ratio; 95% CI: 95% confidence interval.

## Data Availability

The data that support the findings of this study are available from the corresponding author upon request.

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
