# Peer review of "Differential Sensitivity of Germline and Somatic BRCA Variants to PARP Inhibitor in High-Grade Serous Ovarian Cancer"

_ijms, 2023, doi:10.3390/ijms241814181_

Round 1

Reviewer 1 Report

In the manuscript entitled “Differential Sensitivity of Germline and Somatic BRCA Variants to PARP Inhibitor in High-Grade Serous Ovarian Cancer”, Vendrell and colleagues performed a retrospective study of HGSOC tumors diagnosed at their hospital. They identified whether alterations in BRCA1 and BRCA2 were of somatic or germline nature and evaluated how these alterations influenced the response to chemotherapy and PARPi, as well as their efficacy on overall outcomes.

Findings from this work offer a potentially interesting insight into the role of somatic or germline BRCA1/2 mutation as a potential determinant in response to PARP inhibitor. However, the current work is not straightforward enough to support the authors’ conclusions. The authors should still address the following issues and concerns to improve this study to a better level:

1. The resolution of the figures is low. For example, the unclarity of Figure 3B and 3C makes it hard to distinguish the color pattern. Could the authors improve all the figures to make them clearer?

2. The legends of Table 1 are not related. There are no P values, VAF and SNV in that table. Please correct or add related results.

3. There is no citation of Table 2 in the results. What is the relationship between Table 2 and Figure 4? Please explain the results of Table 2 explicitly.

4. In addition to the results in Figure 4, it will be more interesting if the authors could plot the clinical outcomes of patients according to the germline or somatic status separately in BRCA1 or BRCA2 variant types.

5. The authors didn’t discuss the possible reasons for the different clinical outcomes of the germline and somatic mutations. Please add some discussion to such main findings of this paper.

Reviewer 2 Report

no

Reviewer 3 Report

I read with great interest the Manuscript titled " Differential Sensitivity of Germline and Somatic BRCA Variants to PARP Inhibitor in High-Grade Serous Ovarian Cancer”, topic interesting enough to attract readers' attention.

Although the manuscript can be considered already of good quality, I would suggest following recommendations: 

-       I suggest a round of language revision, in order to correct few typos and improve readability.

-       The authors could extend and improve the discussion by evaluating and citing current evidence about possible therapeutic strategies for patients with ovarian cancer.  I would be glad if the authors discuss this important point, referring to PMID: 37314974 and 27794568.

Because of these reasons, the article should be revised and completed. Considering all these points, I think it could be of interest to the readers and, in my opinion, it deserves the priority to be published after minor revisions.

I suggest a round of language revision, in order to correct few typos and improve readability.

Round 2

Reviewer 1 Report

The authors have largely addressed the major concerns and significantly improved the manuscript. I have no hesitation to suggest the publication of this paper.

Reviewer 2 Report

All the comments are addressed nicely.